# Tunable Aryl Imidazolium Recyclable Ionic Liquid with Dual Brønsted–Lewis Acid as Green Catalyst for Friedel–Crafts Acylation and Thioesterification

**DOI:** 10.3390/molecules25020352

**Published:** 2020-01-15

**Authors:** Yi-Jyun Lin, Yao-Peng Wu, Mayur Thul, Ming-Wei Hung, Shih-Huan Chou, Wen-Tin Chen, Wesley Lin, Michelle Lin, Daggula Mallikarjuna Reddy, Hsin-Ru Wu, Wen-Yueh Ho, Shun-Yuan Luo

**Affiliations:** 1Department of Chemistry, National Chung Hsing University, Taichung 402, Taiwan; jiun16899@yahoo.com.tw (Y.-J.L.); grandy.hchs@gmail.com (Y.-P.W.); mayurthul4@gmail.com (M.T.); wei7521@gmail.com (M.-W.H.); mallikarjuna18.d@gmail.com (D.M.R.); 2Department of Chemistry, Taipei American School, Taipei 111, Taiwan; 20ryanc@tas.tw; 3Department of Chemistry, Taichung Municipal Chungming Senior High School, Taichung 403, Taiwan; ting2182180@gmail.com; 4Department of Chemistry, Morrison Academy, Taichung 406, Taiwan; linw@mca.org.tw (W.L.); linm5@mca.org.tw (M.L.); 5Instrumentation Center, MOST, National Tsing Hua University, Hsinchu 30013, Taiwan; hrwu@mx.nthu.edu.tw; 6Department of Cosmetic Science and Institute of Cosmetic Science, Chia Nan University of Pharmacy and Science, Tainan 717, Taiwan

**Keywords:** ionic liquids, Friedel–Crafts acylation, thioesterification, recyclable

## Abstract

Unique tunable aryl imidazolium ionic liquids successfully catalyzed Friedel–Crafts acylation and thioesterification in sealed tubes. These reactions can form a C−C bond and a C−S bond with high atom economy. Ionic liquids exhibited high activity and catalyzed essential reactions with good to excellent yields while retaining their catalytic activities for recycling.

## 1. Introduction

Ionic liquids outperform organic solvents in industrial processes and are considered as an eco-friendly choice to the broad scope of the organic reactions [1,2,3,4,5,6]. They are important green solvents that exhibit high thermal stability, recovery, and recycling. They are also a new class of solvents that are expected to be increasingly used by the chemical industry in the next few years, replacing volatile organic solvents. Owing to their low volatility, non-flammability, and thermal stability, ionic liquids can be applied in many operations. Most ionic liquids, such as imidazolium, comprise an organic or inorganic anion and a quaternary ammonium cation [7,8,9,10]. Because they are hugely tunable and have remarkable properties, they have become a crucial part in synthesis and catalysis. Most of the interest in ionic liquids concentrates on their ability to change considerably the reactivity of dissolved solutes. The properties of ionic liquids have caused them to be identified as designer solvents, including task specific ionic liquids [11,12]. A comprehensive understanding of the physical characteristics of ionic liquids can increase their industrial use [13]. Ionic liquids have been analyzed owing to their many applications in organic synthesis, analytical chemistry, electrochemistry, separation chemistry, separation technology, polymers, fiber optics, pH sensors, and others [14,15].

In general, electronic effects and polarity of solvents play important roles in the outcomes of the product for Friedel–Crafts reaction. The changing the connected carbon and oxygen atoms were accomplished through para selective functionalization of benzoic acid in the presence of palladium catalyst [16,17,18,19]. The aromatic ketones of Friedel–Crafts acylation are a fundamental mediator in a broad range, such as pharmaceutical dyes, fragrances, and agrochemicals [20], and are convenient for use in the synthesis of poly (4-vinyl pyridine)-triflic acid, indium triflate [21,22], perfluoroalkane sulfonic acidic resin is an acid catalyst with catalytic activity for many reactions giving high selectivity. One major drawback of this catalyst is its inefficient swelling by aprotic organic solvents, which generally leads to low reaction rates and others [23]. Iron (III) chloride earns wider acceptance as a useful Lewis acid in Friedel–Crafts acylation [24]. The most exciting feature of our synthesized ionic liquids have an important role in Friedel–Crafts acylation. Previous investigations have reported that ionic liquids exhibit the dual Brønsted and Lewis acidic property, the halogen-free Brønsted–Lewis acidic ionic liquids were synthesized and exploited to catalyze the esterification of caprylic acid with methanol. The novel multifunctional MCM-41 as Brønsted–Lewis acidic ionic liquids were prepared and tested for their catalytic activities in one-pot three-component Mannich reactions [25,26,27]. Benzoylation of anisole catalyzed by metal triflate and chloroindate (III) by Lewis—acid ionic liquid were used in Friedel–Crafts reaction [28,29]. The consolidation of sp^3^ alkyl and sp^2^ aryl substituents at the nitrogen atoms of the imidazolium origin allowed a far greater variety of ionic liquids [30]. These ionic liquids have an sp^2^ hybridized carbon atoms as an N-substituted heterocycle are synthesized as a novel type of ionic liquids is a renowned catalyst which is moisture insensitive and stable at room temperature, use for a variety of organic transformations, including Friedel–Crafts acylation reactions. The electron-withdrawing group of the aryl ring allows easier deprotonation of imidazolium for forming a stronger Brønsted acid.

Thioesters are important molecules for organic synthesis and are obtained by coupling aldehydes and sulfur surrogates recently [31]. Kita and co-workers described the formation of thioesters from aldehydes [32] and specific pentafluorophenyl disulfide [33]. Takemoto and co-workers reported the expensive carbene-promoted coupling reaction of thiols and aldehydes [34]. Bandgar and co-workers developed that Dess–Martin periodinane and NaN_3_ promoted to synthesize thio-esters with aldehydes and aryl thiols [35]. Lee and co-workers demonstrated that FeBr_2_ is able to catalyze synthesis of thioesters from thiols, aldehydes and *tert*-butyl hydro peroxide (TBHP) in water [36].

## 2. Results and Discussion

Initial studies of **2a** and **3a** determined the optimal reaction conditions (Table 1). The study of dual Brønsted–Lewis acidic ionic liquids **1a**, **1b**, **1c**, and **1d** of Figure 1 suggested that **1d** was the best catalyst of Friedel–Crafts acylation, providing the product **4a** in 74% yield (Table 1, entries 1–4) [37,38]. Increasing the temperature to 100 °C and 120 °C with **1d** afforded **4a** in 74% and 71% yields, respectively (Table 1, entries 5–6). Lowering the concentration of **1d** to 0.9 equivalent led to higher yield (78%, Table 1, entry 7), while decreasing it to 0.8 equivalent reduced the reaction yield (73%, Table 1, entry 8).

### 2.1. Friedel–Crafts Acylation

The scope of Friedel–Crafts reaction with various aryl alkanes and acyl chlorides under optimized reaction conditions was investigated (Table 2, entries 1–8). The Friedel–Crafts reactions with aryl alkanes **2a** and different acyl chloride (**3b**–**3c**) under standard conditions were carried out, and the desired products **4b** and **4c** were isolated in 73% and 68% yields (Table 2, entries 1,2). Under the same reaction conditions, the Friedel–Crafts reaction in the presence of **2b** and **3a** produced **4d** in 83% yield (Table 1, entry 3). The reaction proceeded very smoothly with **2b** and **3b** under the above conditions to yield the compound **4e** with a 79% yield (Table 2, entry 4). Acyl chloride **3c** reacted with **2b** to afford the corresponding product **4f** in 89% yield (Table 2, entry 5). Aryl alkane **2c** and acyl chloride **3a** were coupled under similar reaction conditions to obtain the product **4g** in 81% yield (Table 2, entry 6). When the reaction was performed using **2c** and **3b**, the product **4h** was afforded in 71% yield (Table 2, entry 7). Product **4i** was furnished in 70% yield using **2c** and **3c** (Table 2, entry 8). While doing Friedel–Crafts acylation, HCl did not affect the reaction, so could not consider to remove HCl from ionic liquid. It is possible to remove HCl from an ionic liquid by using a high vacuum or suction pump.

To further develop a recyclable catalytic system, the recycling of the **1d** was inspected under the optimized conditions (Figure 2a). After reaction finished, diethyl ether was added. The ionic liquid **1d** was recovered from water layer and dried for the next reaction under vacuum. For the first run, the activity of ionic liquid **1d** remained same yield (78%). In the subsequent second (76%), third (66%), fourth (69%), and fifth (67%) cycles, the desired product was still reached with 100% conversion, when reaction was monitored on TLC plate, the desired product saw without any side product (Figure 2b).

Initially, acyl chloride **2** was abstracted the proton of **1d**. The acyl chloride cation **A** and **1d_a_** transformed to intermediate **B** by the release of **1d_b_**. Then, **1d_b_** captured the proton of intermediate **B** to give intermediate **C**. Subsequently, **C** was transformed into acyl cation **D** by the release of FeCl_4_ anion. Ultimately, the acyl cation **D** reacted with aryl alkane **1** to form the desired product **3** (Figure 3).

### 2.2. Thioesterification

Initially, benzaldehyde **5a** and 1-dodecanethiol **6a** were selected as the model substrates to determine the optimized reaction conditions and the results are summarized in Table 3. We first examined the source of the ionic liquids **1a**–**1d** (0.025 equivalent) in presence of TBHP as a oxidant at 120 °C (Table 3, entries 1–4), and **1d** was the best catalyst of thioesterification providing the product **7a** in 70% yield (Table 3, entries 1–4). Decreasing the temperature to 100 °C and increasing the temperature to 140 °C with **1d** afforded **7a** in similar yields (Table 3, entries 5–6). Decreasing the concentration of **1d** to 0.010 equivalent led to lower yield (68%, Table 1, entry 7), while increasing it to 0.030 equivalent reduced the reaction yield (61%, Table 1, entry 8).

With these optimized reaction conditions in hand, the scope of the substrates was then studied. The results are summarized in Table 4. A variety of alkyl thiols and aromatic thiols were conducted with aromatic and alkyl aldehydes to afford the corresponding thioesters in good to excellent yields. Aromatic aldehydes bearing electron-donating and electron-withdrawing substituents are all suitable for catalysis. Aldehyde reacted with thiol **6b** and **6c** to give corresponding product **7b** in 88% and **7c** in 55% respectively. It is important to note that this system shows good functional group tolerance; chloro (Table 4, products **7d**, **7e** and **7f**) are tolerated by reaction condition employed. Aldehydes bearing electron donating substituents underwent thioesterification with alkyl and aryl thiols to give desired product (Table 4, products **7g** and **7h**). Alkyl thiols were also reacted with alkyl aldehydes to give the corresponding thioesters (Table 4, products **7i**, **7j**, **7k**, and **7l**).

In this method, the recovery of ionic liquid **1d** ranges from 63–70% yields. The catalytic activity was examined by thioesterification of thiols at 120 °C for 1 h (Figure 4a). The results are presented in (Figure 4b). Ionic liquid was recovered and reused up to five times with only slightly decreased catalytic activity.

Based on the above experimental results, a plausible mechanism is proposed in Figure 5. When **1d** is reacted with TBHP, and t-BuOO^·^ radical is generated at the time. Initially benzaldehyde reacted with ionic liquid **1d**, which would lead to intermediate **3**. Then thiols react with intermediate **3** to give **4**. A hydrogen atom is then abstracted from aldehyde to give an acyl radical. Further t-BuOO^·^ radical reacted with **4** to give complex **5** and then hydroxyl radicle II Reacts with **5** to give the final product **6**.

## 3. Materials and Methods 

The reactions were conducted in flame-dried glassware, under the nitrogen atmosphere. Acetonitrile and dichloromethane were purified and dried from a safe purification system containing activated Al_2_O_3_. All reagents obtained from commercial sources were used without purification unless otherwise mentioned. Flash column chromatography was carried out on Silica Gel 60. TLC was performed on pre-coated glass plates of Silica Gel 60 F254 detection was executed by spraying with a solution of Ce(NH_4_)_2_(NO_3_)_6_ (0.5 g), (NH_4_)_6_Mo_7_O_24_ (24.0 g), and H_2_SO_4_ (28.0 mL) in water (500.0 mL) and subsequent heating on a hot plate. Optical rotations were measured at 589 nm (Na), ^1^H, ^13^C NMR, DEPT, ^1^H-^1^H COSY, ^1^H-^13^C COSY, and NOESY spectra were recorded with 400 MHz instruments. Chemical shifts are in ppm from Me_4_Si generated from the CDCl_3_ lock signal at δ 7.26. IR spectra were taken with a FT-IR spectrometer using NaCl plates. Mass spectra were analyzed on instrument with an EI, ESI, APCI, and FAB source.

*(4-Methoxyphenyl)-phenylmethanone* (**4a**). To a solution of aryl alkane **2a** (217 μL, 2 mmol), acyl chloride **3a** (116 μL, 1 mmol), and ionic liquid **1d** (349 mg, 0.9 mmol) were stirred at 100 °C for two hours. After cooling, the reaction mixture was washed by diethyl ether (2 × 40 mL). The diethyl ether layer was decanted, extracted with water, aqueous NaHCO_3_, and brine, and dried over MgSO_4_. After filtration, the organic solvent was then removed on a rotary evaporator. The residue was purified by flash chromatography to give the desired product 4a (165 mg, 78%) as a light yellow oil. R_f_ 0.37 (EtOAc/Hex = 1/6). IR (NaCl) v 3060, 3006, 2840, 1651, 1597, 1508, 1171 cm^−1^; ^1^H NMR (Appendix A) (400 MHz, CDCl_3_) δ 7.82 (d, *J* = 8.8 Hz, 2H), 7.74 (d, *J* = 6.8 Hz, 2H), 7.55 (t, *J* = 7.4 Hz, 1H), 7.45 (t, *J* = 7.6 Hz, 2H), 6.95 (d, *J* = 9.2 Hz, 2H), 3.86 (s, 3H); ^13^C NMR (101 MHz, CDCl_3_) δ 195.4, 163.1, 138.2, 132.4, 131.8, 130.0, 129.6, 128.1, 113.4, 55.4; HRMS (EI, M^+^) calculated for C_14_H_12_O_2_ 212.0837, found 212.0834.

*(2-Chlorophenyl)-(4-methoxyphenyl) methanone* (**4b**). To a solution of aryl alkane **2a** (217 μL, 2 mmol), acyl chloride **3b** (126 μL, 1 mmol), and ionic liquid **1d** (349 mg, 0.9 mmol) were stirred at 100 °C for four hours. After cooling, the reaction mixture was washed by diethyl ether (2 × 40 mL). The diethyl ether layer was decanted, extracted with water, aqueous NaHCO_3_, and brine, and dried over MgSO_4_. After filtration, the organic solvent was removed on a rotary evaporator. The residue was purified by flash chromatography to give the desired product **4b** (180 mg, 73%) as a light yellow solid. R_f_ 0.40 (EtOAc/Hex = 1/4); mp 71–74 °C; IR (NaCl) v 3068, 3009, 2964, 2840, 1659, 1595, 1508, 1464, 1149, 843 cm^−1^; ^1^H NMR (400 MHz, CDCl_3_) δ 7.79 (d, *J* = 9.2 Hz, 2H), 7.47–7.38 (m, 2H), 7.37–7.36 (m, 1H), 7.35 (d, *J* = 1.2 Hz, 1H), 6.94 (d, *J* = 9.2 Hz, 2H), 3.88 (s, 3H); ^13^C NMR (101 MHz, CDCl_3_) δ 193.8, 164.1, 139.0, 132.5, 131.0, 130.8, 129.9, 129.4, 128.8, 126.6, 113.8, 55.5; HRMS (ESI, M + H^+^) calculated for C_14_H_12_ClO_2_ 247.0526, found 247.0525

*(4-Chlorophenyl)-(4-methoxyphenyl)methanone* (**4c**). To a solution of aryl alkane **2a** (217 μL, 2 mmol), acyl chloride **3c** (127 μL, 1 mmol), and ionic liquid **1d** (349 mg, 0.9 mmol) were stirred at 100 °C for four hours. After cooling, the reaction mixture was washed by diethyl ether (2 × 40 mL). The diethyl ether layer was decanted, extracted with water, aqueous NaHCO_3_, and brine, and dried over MgSO_4_. After filtration, the organic solvent was removed on a rotary evaporator. The residue was purified by flash chromatography to give the desired product **4c** (168 mg, 68%) as a light yellow solid. R_f_ 0.37 (EtOAc/Hex = 1/4); mp 119–122 °C; IR (NaCl) v 2962, 2934, 2841, 1641, 1604, 1509, 1461, 1148, 760 cm^−1^; ^1^H NMR (400 MHz, CDCl_3_) δ 7.80 (d, *J* = 8.4 Hz, 2H), 7.71 (d, *J* = 8.4 Hz, 2H), 7.45 (d, *J* = 8.4 Hz, 2H), 6.97 (d, *J* = 8.8 Hz, 2H), 3.89 (s, 3H); ^13^C NMR (101 MHz, CDCl_3_) δ 194.2, 163.3, 138.2, 136.5, 132.4, 131.1, 129.7, 128.5, 113.6, 55.5; HRMS (ESI, M + H^+^) calculated for C_14_H_12_ClO_2_ 247.0526, found 247.0523.

*(2,4,6-Trimethylphenyl)-phenylmethanone* (**4d**). To a solution of aryl alkane **2b** (278 µL, 2 mmol), acyl chloride **3a** (116 µL, 1 mmol) and ionic liquid 5 (349 mg, 0.9 mmol) were stirred at 100 °C for 2 hours. After cooling, the reaction mixture was washed by diethyl ether (2 × 40 mL). The diethyl ether layer was decanted, extracted with water, aqueous NaHCO_3_, and brine, and dried over MgSO_4_. After filtration, the organic solvent was then removed on a rotary evaporator. The residue was purified by flash chromatography to give the desired product **4d** (186 mg, 83%) as a light yellow oil. R_f_ 0.40 (EtOAc/Hex = 1/12); IR (NaCl) v 3061, 2951, 2921, 2860, 1671, 1449, 1380 cm^−1^; ^1^H NMR (400 MHz, CDCl_3_) δ 7.81 (d, *J* = 7.2 Hz, 2H), 7.57 (t, *J* = 7.2 Hz, 1H), 7.44 (t, *J* = 7.8 Hz, 2H), 6.90 (s, 2H), 2.34 (s, 3H), 2.09 (s, 6H); ^13^C NMR (101 MHz, CDCl_3_) δ 200.8, 138.5, 137.3, 136.8, 134.1, 133.5, 129.4, 128.7, 128.3, 21.1, 19.3; HRMS (ESI, M + H^+^) calculated for C_16_H_17_O 225.1279, found 225.1281.

*(2-Chlorophenyl)-(2,4,6-trimethylphenyl) methanone* (**4e**). To a solution of aryl alkane **2b** (278 μL, 2 mmol), acyl chloride **3b** (126 μL, 1 mmol), and ionic liquid **1d** (349 mg, 0.9 mmol) were stirred at 100 °C for 3.5 hours. After cooling, the reaction mixture was washed by diethyl ether (2 × 40 mL). The diethyl ether layer was decanted, extracted with water, aqueous NaHCO_3_, and brine, and dried over MgSO_4_. After filtration, the organic solvent was then removed on a rotary evaporator. The residue was purified by flash chromatography to give the desired product **4e** (204 mg, 79%) as a light orange solid. R_f_ 0.53 (EtOAc/Hex = 1/10). mp 101–102 °C; IR (NaCl) v 2917, 1671, 1610, 1584, 1436 cm^−1^; ^1^H NMR (400 MHz, CDCl_3_) δ 7.47 (d, *J* = 7.8 Hz, 2H), 7.41 (t, *J* = 7.7 Hz, 1H), 7.26 (t, *J* = 7.5 Hz, 1H), 6.87 (s, 2H), 2.31 (s, 3H), 2.12 (s, 6H); ^13^C NMR (101 MHz, CDCl_3_) δ 198.9, 139.3, 137.6, 137.3, 135.0, 133.2, 132.7, 131.7, 131.4, 128.8, 126.8, 21.1, 19.7; HRMS (ESI, M + Na^+^) calculated for C_16_H_15_ClONa 281.0709, found 281.0710.

*(4-Chlorophenyl)-(2,4,6-trimethylphenyl)methanone* (**4f**). To a solution of aryl alkane **2b** (278 μL, 2 mmol), acyl chloride **3c** (127 μL, 1 mmol), and ionic liquid **1d** (349 mg, 0.9 mmol) were stirred at 100 °C for two hours. After cooling, the reaction mixture was washed by diethyl ether (2 × 40 mL). The diethyl ether layer was decanted, extracted with water, aqueous NaHCO_3_, and brine, and dried over MgSO_4_. After filtration, the organic solvent was then removed on a rotary evaporator. The residue was purified by flash chromatography to give the desired product **4f** (230 mg, 89%) as a white solid. R_f_ 0.61 (EtOAc/Hex = 1/10). mp 64–65 °C; IR (NaCl) v 2921, 1673, 1586 cm^−1^; ^1^H NMR (400 MHz, CDCl_3_) δ 7.74 (d, *J* = 8.4 Hz, 2H), 7.41 (d, *J* = 8.5 Hz, 2H), 6.90 (s, 2H), 2.33 (s, 3H), 2.07 (s, 6H); ^13^C NMR (101 MHz, CDCl_3_) δ 199.5, 140.1, 138.7, 136.3, 135.6, 134.1, 130.7, 129.1, 128.4, 21.1, 19.3; HRMS (APCI, M + H^+^) calculated for C_16_H_16_ClO 259.0890, found 259.0888.

*(3,4-Dimethoxyphenyl)-phenylmethanone* (**4g**). To a solution of aryl alkane **2c** (256 µL, 2 mmol), acyl chloride **3a** (116 µL, 1 mmol) and ionic liquid **1d** (349 mg, 0.9 mmol) were stirred at 100 °C for three hours. After cooling, the reaction mixture was washed by diethyl ether (2 × 40 mL). The diethyl ether layer was decanted, extracted with water, aqueous NaHCO_3_, and brine, and dried over MgSO_4_. After filtration, the organic solvent was then removed on a rotary evaporator. The residue was purified by flash chromatography to give the desired product **4g** (196 mg, 81%) as a white solid. R_f_ 0.45 (EtOAc/Hex = 1/4); mp 100–101 °C; IR (NaCl) v 3079, 3003, 2960, 2839, 1649, 1594, 1272, 1130 cm^−1^; ^1^H NMR (400 MHz, CDCl_3_) δ 7.75 (d, *J* = 8.2 Hz, 2H), 7.55 (t, *J* = 7.4 Hz, 1H), 7.48 (d, *J* = 2.0 Hz, 1H), 7.46 (t, *J* = 7.5 Hz, 2H), 7.36 (dd, *J* = 8.3, 2.0 Hz, 1H), 6.88 (d, *J* = 8.4 Hz, 1H), 3.94 (s, 3H), 3.92 (s, 3H); ^13^C NMR (101 MHz, CDCl_3_) δ 195.4, 152.9, 148.9, 138.1, 131.7, 130.0, 129.6, 128.0, 125.4, 111.9, 109.6, 55.9; HRMS (ESI, M + Na^+^) calculated for C_15_H_14_O_3_Na 265.0841, found 265.0844.

*(2-Chlorophenyl)-(3,4-dimethoxyphenyl) methanone* (**4h**). To a solution of aryl alkane **2c** (256 μL, 2 mmol), acyl chloride **3b** (126 μL, 1 mmol), and ionic liquid **1d** (349 mg, 0.9 mmol) were stirred at 100 °C for 3.5 hours. After cooling, the reaction mixture was washed by diethyl ether (2 × 40 mL). The diethyl ether layer was decanted, extracted with water, aqueous NaHCO_3_, and brine, and dried over MgSO_4_. After filtration, the organic solvent was then removed on a rotary evaporator. The residue was purified by flash chromatography to give the desired product **4h** (196 mg, 71%) as a light yellow solid. R_f_ 0.34 (EtOAc/Hex = 1/2). mp 142–143 °C; IR (NaCl) v 3079, 2936, 2840, 1659, 1592, 1513, 1464, 1418, 1133 cm^−1^; ^1^H NMR (400 MHz, CDCl_3_) δ 7.56 (d, *J* = 2.0 Hz, 1H), 7.45–7.37 (m, 2H), 7.34 (t, *J* = 1.0 Hz, 1H), 7.33 (d, *J* = 1.1 Hz, 1H), 7.19 (dd, *J* = 8.4, 2.0 Hz, 1H), 6.82 (d, *J* = 8.4 Hz, 1H), 3.92 (d, *J* = 1.8 Hz, 6H); ^13^C NMR (101 MHz, CDCl_3_) δ 193.8, 153.9, 149.2, 138.8, 131.0, 130.7, 129.9, 129.4, 128.8, 126.5, 126.3, 110.7, 109.9, 56.1, 55.9; HRMS (ESI, M + H^+^) calculated for C_15_H_14_ClO_3_ 277.0631, found 277.0632.

*(4-Chlorophenyl)-(3,4-dimethoxyphenyl)methanone* (**4i**). To a solution of aryl alkane **2c** (256 µL, 2 mmol), acyl chloride **3c** (127 µL, 1 mmol), and ionic liquid **1d** (349 mg, 0.9 mmol) were stirred at 100 °C for three hours. After cooling, the reaction mixture was extracted by ethyl acetate (2 × 40 mL). The ethyl acetate layer was decanted, washed with water, aqueous NaHCO_3_, and brine, and dried over MgSO_4_. The residue was purified by flash chromatography to give the desired product **4i** (193 mg, 70%) as a white solid. R_f_ 0.28 (EtOAc/Hex = 1/4). mp 113–114 °C; IR (NaCl) v 2935, 2839, 1649, 1594, 1514, 1272 cm^−1^; ^1^H NMR (400 MHz, CDCl_3_) δ 7.72 (d, *J* = 8.4 Hz, 2H), 7.48–7.46 (m, 2H), 7.45–7.44 (m, 1H), 7.34 (dd, *J* = 8.0, 2.0 Hz, 1H), 6.90 (d, *J* = 8.4 Hz, 1H), 3.97 (s, 3H), 3.95 (s, 3H); ^13^C NMR (101 MHz, CDCl_3_) δ 193.8, 152.9, 148.8, 137.9, 136.2, 130.9, 129.5, 128.2, 125.1, 111.6, 109.5, 55.8, 55.7; HRMS (ESI, M + H^+^) calculated for C_15_H_14_ClO_3_ 277.0632, found 277.0656.

*S-Dodecyl benzothioate* (**7a**). To a solution of thiol **6a** (240 µL, 1 mmol), aldehyde **5a** (510 µL, 5 mmol), tert-butyl hydroperoxide (277 µL, 2 mmol), and ionic liquid **1d** (9.7 mg, 0.025 mmol) were stirred at 120 °C for one hour in a sealed tube. After cooling, the reaction mixture was extracted with ethyl acetate (3 × 20 mL). The combined organic layers were dried over anhydrous MgSO_4_, filtered and concentrated. The residue was purified by flash chromatography to give the desire product **7a** (214 mg, 70% yield) as a colorless liquid. R_f_ 0.36 (Hexane). IR (NaCl) v 3063, 3030, 2925, 2854, 1666, 1597, 1449, 1377, 1027, 1001 cm^−1^; ^1^H NMR (400 MHz, CDCl_3_) δ 7.97 (dd, *J* = 8.4, 1.2 Hz, 2H), 7.52 (t, *J* = 7.4 Hz, 1H), 7.41 (t, *J* = 7.6 Hz, 2H), 3.06 (t, *J* = 7.4 Hz, 2H), 1.72–1.62 (m, 2H), 1.46–1.38 (m, 2H), 1.27 (s, 16H), 0.89 (t, *J* = 7.0 Hz, 3H); ^13^C NMR (101 MHz, CDCl_3_) δ 191.7, 137.1, 133.0, 128.4, 127.0, 31.9, 29.6, 29.5, 29.4, 29.3, 29.1, 28.9, 28.88, 22.6, 14.0; HRMS (FAB, M + H^+^) calculated for C_19_H_31_OS 307.2096, found 307.2087.

*S-Decyl benzothioate* (**7b**). To a solution of thiol **6b** (212 µL, 1 mmol), aldehyde **5a** (510 µL, 5 mmol), tert-butyl hydroperoxide (277 µL, 2 mmol), and ionic liquid **1d** (9.7 mg, 0.025 mmol) were stirred at 120 °C for 1 hour in a sealed tube. After cooling, the reaction mixture was extracted with ethyl acetate (3 × 20 mL). The combined organic layers were dried over anhydrous MgSO_4_, filtered and concentrated. The residue was purified by flash chromatography to give the desire product **7b** (245 mg, 88% yield) as a colorless liquid. R_f_ 0.30 (Hexane). IR (NaCl) v 3063, 3031, 2926, 2854, 1666, 1597, 1449, 1377, 1027, 1001 cm^−1^; ^1^H NMR (400 MHz, CDCl_3_) δ 7.97 (d, *J* = 7.2 Hz, 2H), 7.53 (t, *J* = 7.4 Hz, 1H), 7.42 (t, *J* = 7.8 Hz, 2H), 3.06 (t, *J* = 7.4 Hz, 2H), 1.63–1.71 (m, 2H), 1.46–1.38 (m, 2H), 1.27 (s, 12H), 0.89 (t, *J* = 6.8 Hz, 3H); ^13^C NMR (101 MHz, CDCl_3_) δ 191.9, 137.2, 133.0, 128.4, 127.1, 31.8, 29.5, 29.49, 29.45, 29.2, 29.1, 29.0, 28.9, 22.6, 14.0; HRMS (ESI, M + Na^+^) calculated for C_17_H_26_NaOS 301.1602, found 301.1606.

*S-(4-Chlorophenyl) benzothioate* (**7c**). To a solution of thiol **6c** (145 mg, 1 mmol), aldehyde **5a** (510 µL, 5 mmol), tert-butyl hydroperoxide (277 µL, 2 mmol), and ionic liquid **1d** (9.7 mg, 0.025 mmol) were stirred at 120 °C for one hour in a sealed tube. After cooling, the reaction mixture was extracted with ethyl acetate (3 × 20 mL). The combined organic layers were dried over anhydrous MgSO_4_, filtered and concentrated. The residue was purified by flash chromatography to give the desire product **7c** (137 mg, 55% yield) as a white solid. R_f_ 0.25 (Hexane). mp 75–76 °C; IR (NaCl) v 3082, 3055, 1674, 1574, 1474, 1446, 1389, 1203, 1012 cm^−1^; ^1^H NMR (400 MHz, CDCl_3_) δ 8.03 (dd, *J* = 8.4, 1.2 Hz, 2H), 7.62 (t, *J* = 7.4 Hz, 1H), 7.50 (t, *J* = 7.8 Hz, 2H), 7.47–7.41 (m, 4H); ^13^C NMR (101 MHz, CDCl_3_) δ 189.5, 136.2, 135.9, 133.8, 129.4, 128.7, 127.4, 125.7; HRMS (ESI, M + Na^+^) calculated for C_13_H_9_ClNaOS 270.9960, found 270.9965.

*S-Dodecyl 4-chlorobenzothioate* (**7d**). To a solution of thiol **6a** (240 µL, 1 mmol), aldehyde **5b** (703 µL, 5 mmol), tert-butyl hydroperoxide (277 µL, 2 mmol), and ionic liquid **1d** (9.7 mg, 0.025 mmol) were stirred at 120 °C for one hour in a sealed tube. After cooling, the reaction mixture was extracted with ethyl acetate (3 × 20 mL). The combined organic layers were dried over anhydrous MgSO_4_, filtered, and concentrated. The residue was purified by flash chromatography to give the desire product **7d** (248 mg, 73%) as a colorless liquid. R_f_ 0.28 (Hexane). IR (NaCl) v 2925, 2854, 1913, 1825, 1785, 1668, 1589, 1464, 1092 cm^1^; ^1^H NMR (400 MHz, CDCl_3_) δ 7.90 (d, *J* = 8.8 Hz, 2H), 7.41 (d, *J* = 8.8 Hz, 2H), 3.07 (t, *J* = 7.4 Hz, 2H), 1.67 (quint, *J* = 7.4 Hz, 2H), 1.42 (quint, *J* = 7.2 Hz, 2H), 1.26 (s, 16H), 0.88 (t, *J* = 7.0 Hz, 3H); ^13^C NMR (101 MHz, CDCl_3_) δ 190.9, 139.5, 135.5, 128.8, 128.5, 31.9, 29.6, 29.56, 29.5, 29.3, 29.2, 29.1, 28.9, 22.7, 14.1; HRMS (ESI, M + H^+^) calculated for C_19_H_30_ClOS 341.1706, found 341.1705.

*S-Decyl 4-chlorobenzothioate* (**7e**). To a solution of thiol **6b** (221 µL, 1 mmol), aldehyde **5b** (703 mg, 5 mmol), tert-butyl hydroperoxide (277 µL, 2 mmol), and ionic liquid **1d** (9.7 mg, 0.025 mmol) were stirred at 120 °C for one hour in a sealed tube. After cooling, the reaction mixture was extracted with ethyl acetate (3 × 20 mL). The combined organic layers were dried over anhydrous MgSO_4_, filtered, and concentrated. The residue was purified by flash chromatography to give the desire product **7e** (228 mg, 73%) as a colorless liquid. R_f_ 0.31 (Hexane). IR (NaCl) v 2955, 2926, 2854, 1668, 1589, 1464, 1092 cm^−1^; ^1^H NMR (400 MHz, CDCl_3_) δ 7.90 (d, *J* = 8.4 Hz, 2H), 7.41 (d, *J* = 8.4 Hz, 2H), 3.06 (t, *J* = 7.2 Hz, 2H), 1.66 (quint, *J* = 7.4 Hz, 2H), 1.41 (quint, *J* = 7.2 Hz, 2H), 1.26 (s, 12H), 0.88 (t, *J* = 6.8 Hz, 3H); ^13^C NMR (101 MHz, CDCl_3_) δ 191.0, 139.5, 135.6, 128.8, 128.5, 31.9, 29.5, 29.47, 29.3, 29.2, 29.1, 28.9, 22.7, 14.1; HRMS (ESI, M + H^+^) calculated for C_17_H_26_ClOS 313.1393, found 313.1355.

*S-Dodecyl 4-methoxybenzothioate* (**7f**). To a solution of thiol **6a** (239 μL, 1 mmol), aldehyde **5c** (609 μL, 5 mmol), tert-butyl hydroperoxide (276 μL, 2 mmol), and ionic liquid **1d** (10 mg, 0.025 mmol) were stirred at 140 °C for one hour in a sealed tube. After cooling, the reaction mixture was extracted with ethyl acetate (3 × 20 mL). The combined organic layers were dried over anhydrous MgSO_4_, filtered, and concentrated. The residue was purified by flash chromatography to give the desire product **7f** (255 mg, 76%) as a colorless liquid. R_f_ 0.20 (hexane); IR (NaCl) v 2924, 2853, 1658, 1602, 1508, 1463, 1259, 837; ^1^H NMR (400 MHz, CDCl_3_) δ 7.93 (d, *J* = 9.2 Hz, 2H), 6.89 (d, *J* = 8.8 Hz, 2H), 3.83 (s, 3H), 3.03 (t, *J* = 7.4 Hz, 2H), 1.68–1.60 (m, 2H), 1.44–1.36 (m, 2H), 1.24 (s, 16H), 0.86 (t, *J* = 6.8 Hz, 3H); ^13^C NMR (100 MHz, CDCl_3_) δ 190.4, 163.5, 130.0, 129.2, 113.5, 55.3, 31.9, 29.6, 29.58, 29.54, 29.5, 29.3, 29.1, 28.9, 28.8, 22.6, 14.0; HRMS (ESI, M + Na^+^) calculated for C_20_H_32_NaO_2_S 359.2020, found 359.2032.

*S-Decyl 4-methoxybenzothioate* (**7g**). To a solution of thiol **6b** (207 μL, 1 mmol), aldehyde **5c** (609 μL, 5 mmol), tert-butyl hydroperoxide (276 μL, 2 mmol), and ionic liquid **1d** (10 mg, 0.025 mmol) were stirred at 120 °C for one hour in a sealed tube. After cooling, the reaction mixture was extracted with ethyl acetate (3 × 20 mL). The combined organic layers were dried over anhydrous MgSO_4_, filtered, and concentrated. The residue was purified by flash chromatography to give the desire product **7g** (226 mg, 73%) as a colorless liquid. R_f_ 0.25 (hexane); IR (NaCl) v 2925, 2854, 1658, 1602, 1508, 1463, 1259, 838; ^1^H NMR (400 MHz, CDCl_3_) δ 7.93 (d, *J* = 8.8 Hz, 2H), 6.90 (d, *J* = 8.8 Hz, 2H), 3.84 (s, 3H), 3.02 (t, *J* = 7.4 Hz, 2H), 1.68–1.60 (m, 2H), 1.44–1.35 (m, 2H), 1.24 (s, 12H), 0.86 (t, *J* = 6.8 Hz, 3H); ^13^C NMR (100 MHz, CDCl_3_) δ 190.4, 163.5, 130.0, 129.2, 113.6, 55.3, 31.8, 29.6, 29.5, 29.4, 29.2, 29.1, 28.9, 28.8, 22.6, 14.0; HRMS (ESI, M + Na^+^) calculated for C_18_H_28_NaO_2_S 331.1708, found 331.1702.

*S-(4-Chlorophenyl) 4-methoxybenzothioate* (**7h**). To a solution of thiol **6c** (145 mg, 1 mmol), aldehyde **5c** (609 μL, 5 mmol), tert-butyl hydroperoxide (276 μL, 2 mmol), and ionic liquid **1d** (10 mg, 0.025 mmol) were stirred at 120 °C for one hour in a sealed tube. After cooling, the reaction mixture was extracted with ethyl acetate (3 × 20 mL). The combined organic layers were dried over anhydrous MgSO_4_, filtered and concentrated. The residue was purified by flash chromatography to give the desire product **7h** (142 mg, 51%) as a white solid. R_f_ 0.45 (EtOAc /Hex = 1/4); mp 96–97 °C; IR (NaCl) v 1690, 1660, 1602, 1508, 1261, 833; ^1^H NMR (400 MHz, CDCl_3_) δ 7.98 (d, *J* = 9.2 Hz, 2H), 7.41 (d, *J* = 3.2 Hz, 4H), 6.94 (d, *J* = 8.8 Hz, 2H), 3.84 (s, 3H).; ^13^C NMR (100 MHz, CDCl_3_) δ 187.7, 164.0, 136.2, 135.6, 129.6, 129.2, 128.9, 126.0, 113.8, 55.4; HRMS (ESI, M + Na^+^) calculated for C_14_H_11_ClNaO_2_S 301.0066, found 301.0059.

*S-Dodecyl hexanethioate* (**7i**). To a solution of thiol **6a** (239 μL, 1 mmol), aldehyde **5d** (615 μL, 5 mmol), tert-butyl hydroperoxide (276 μL, 2 mmol), and ionic liquid **1d** (10 mg, 0.025 mmol) were stirred at 120 °C for one hour in a sealed tube. After cooling, the reaction mixture was extracted with ethyl acetate (3 × 20 mL). The combined organic layers were dried over anhydrous MgSO_4_, filtered, and concentrated. The residue was purified by flash chromatography to give the desire product **7i** (240 mg, 80%) as a colorless liquid. R_f_ 0.35 (hexane); IR (NaCl) v 2957, 2926, 2855, 1693, 1464, 1121; ^1^H NMR (400 MHz, CDCl_3_) δ 2.85 (t, *J* = 7.4 Hz, 2H), 2.52 (t, *J* = 7.6 Hz, 2H), 1.69–1.61 (m, 2H), 1.59–1.51 (m, 2H), 1.37–1.27 (m, 8H), 1.25 (s, 14H), 0.90–0.85 (m, 6H); ^13^C NMR (100 MHz, CDCl_3_) δ 199.5, 199.3, 44.0, 31.9, 31.0, 29.6, 29.5, 29.4, 29.3, 29.1, 28.8, 28.7, 25.3, 22.6, 22.3, 14.0, 13.8; HRMS (FAB, M + H^+^) calculated for C_18_H_37_OS 301.2565, found 301.2568.

*S-Decyl hexanethioate* (**7j**). To a solution of thiol **6b** (221 µL, 1 mmol), aldehyde **5d** (615 µL, 5 mmol), tert-butyl hydroperoxide (277 µL, 2 mmol), and ionic liquid **1d** (9.7 mg, 0.025 mmol) were stirred at 120 °C for one hour in a sealed tube. After cooling, the reaction mixture was extracted with ethyl acetate (3 × 20 mL). The combined organic layers were dried over anhydrous MgSO_4_, filtered, and concentrated. The residue was purified by flash chromatography to give the desire product **7j** (194 mg, 71%) as a colorless liquid. R_f_ 0.40 (Hexane). IR (NaCl) v 2927, 2855, 2731, 2671, 1693, 1463, 1030 cm^−1^; ^1^H NMR (400 MHz, CDCl_3_) δ 2.85 (t, *J* = 7.4 Hz, 2H), 2.52 (t, *J* = 7.6 Hz, 2H), 1.65 (quint, *J* = 7.4 Hz, 2H), 1.55 (quint, *J* = 7.3 Hz, 2H), 1.30 (m, 8H), 1.25 (s, 10H), 0.88 (q, *J* = 6.4 Hz, 6H); ^13^C NMR (101 MHz, CDCl_3_) δ 199.9, 44.1, 31.9, 31.1, 29.6, 29.5, 29.48, 29.3, 29.1, 28.82, 28.8, 25.4, 22.7, 22.3, 14.1, 13.9; HRMS (ESI, M + Na^+^) calculated for C_16_H_32_NaOS 295.2072, found 295.2072.

*S-Dodecyl octanethioate* (**7k**). To a solution of thiol **6a** (240 µL, 1 mmol), aldehyde **5e** (781 µL, 5 mmol), tert-butyl hydroperoxide (277 µL, 2 mmol), and ionic liquid **1d** (9.7 mg, 0.025 mmol) were stirred at 120 °C for one hour in a sealed tube. After cooling, the reaction mixture was extracted with ethyl acetate (3 × 20 mL). The combined organic layers were dried over anhydrous MgSO_4_, filtered, and concentrated. The residue was purified by flash chromatography to give the desire product **7k** (223 mg, 68% yield) as a colorless liquid. R_f_ 0.48 (Hexane). IR (NaCl) v 2923, 2855, 1694, 1463, 1377, 1043 cm^−1^; ^1^H NMR (400 MHz, CDCl_3_) δ 2.83 (t, *J* = 7.2 Hz, 2H), 2.50 (t, *J* = 7.4 Hz, 2H), 1.69–1.58 (m, 2H), 1.58–1.48 (m, 2H), 1.23 (s, 26H), 0.85 (t, *J* = 6.6 Hz, 6H); ^13^C NMR (101 MHz, CDCl_3_) δ 199.6, 44.1, 31.9, 31.6, 29.6, 29.58, 29.55, 29.5, 29.3, 29.1, 28.9, 28.88, 28.8, 28.7, 25.7, 22.7, 22.6, 14.1, 14.0; HRMS (FAB, M + H^+^) calculated for C_20_H_41_OS 329.2878, found 329.2878.

*S-Decyl octanethioate* (**7l**). To a solution of thiol **6b** (221 µL, 1 mmol), aldehyde **5e** (781 µL, 5 mmol), tert-butyl hydroperoxide (277 µL, 2 mmol), and ionic liquid **1d** (9.7 mg, 0.025 mmol) were stirred at 120 °C for one hour in a sealed tube. After cooling, the reaction mixture was extracted with ethyl acetate (3 × 20 mL). The combined organic layers were dried over anhydrous MgSO_4_, filtered, and concentrated. The residue was purified by flash chromatography to give the desire product **7l** (192 mg, 64%) as a colorless liquid. R_f_ 0.35 (Hexane). IR (NaCl) v 2956, 2926, 2855, 2730, 2671, 1693, 1464, 1124 cm^−1^; ^1^H NMR (400 MHz, CDCl_3_) δ 2.85 (t, *J* = 7.2 Hz, 2H), 2.52 (t, *J* = 7.4 Hz, 2H), 1.64 (quint, *J* = 7.2 Hz, 2H), 1.55 (quint, *J* = 7.3 Hz, 2H), 1.36–1.26 (m, 10H), 1.25 (s, 12H), 0.87 (t, *J* = 7.0 Hz, 6H); ^13^C NMR (101 MHz, CDCl_3_) δ 199.8, 44.1, 31.9, 31.6, 29.6, 29.5, 29.48, 29.3, 29.1, 28.9, 28.8, 28.79, 25.7, 22.7, 22.6, 14.1, 14.0; HRMS (ESI, M + Na^+^) calculated for C_18_H_36_NaOS 323.2385, found 323.2373.

## 4. Conclusions

The designed ionic liquid **1d** were successfully catalyze the Friedel–Crafts acylation reaction and thioesterification reaction. It provides good to excellent yield in both the reactions under optimal conditions. The ionic liquid **1d** exhibits the dual Brønsted and Lewis acidic property. The catalyst showed high atom economy, high thermal stability, and could be recycled with minor loss in activity and also moisture insensitive. The catalyst shows some limitations, which exhibited good solubility in many organic solvents and deionized water, but not in hexane.

## Figures and Tables

**Figure 1 molecules-25-00352-f001:**
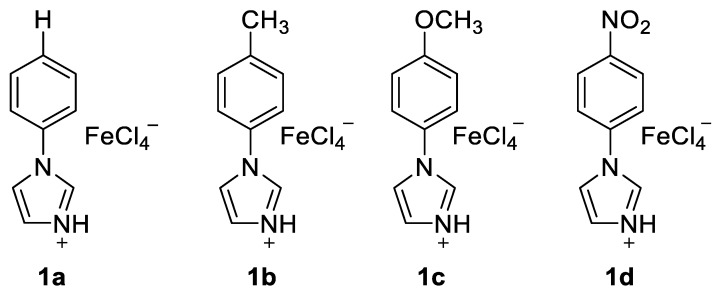
Tunable aryl imidazolium recyclable ionic liquid **1a**–**1d** with dual activity.

**Figure 2 molecules-25-00352-f002:**
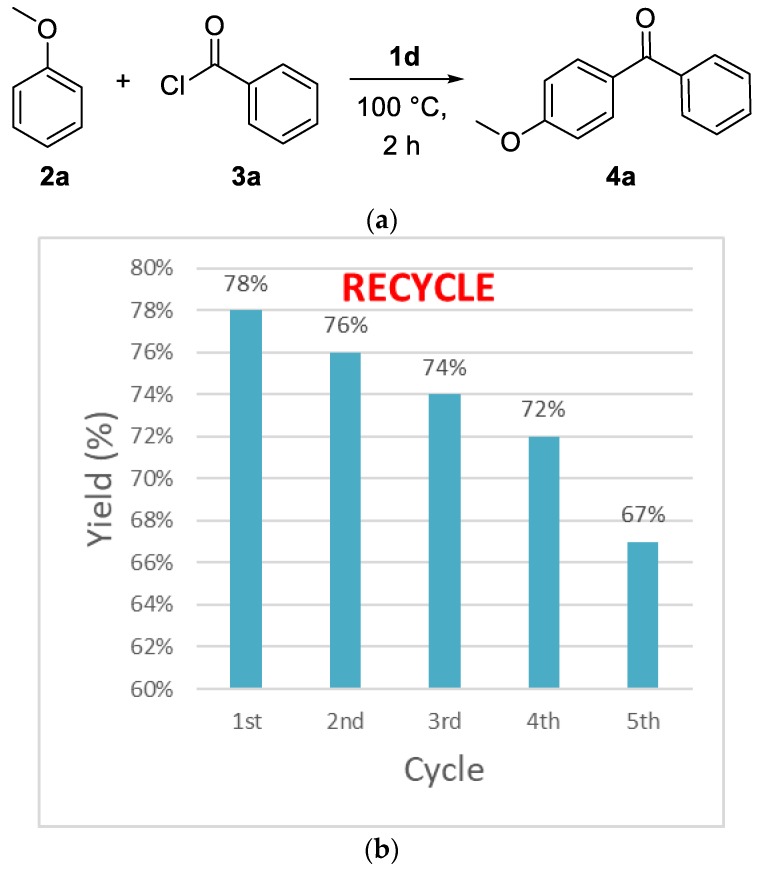
(**a**) Friedel-Crafts acylation reaction (**b**) Recycling of **1d** in the synthesis of (4-methoxyphenyl) phenyl-methanone **4a**.

**Figure 3 molecules-25-00352-f003:**
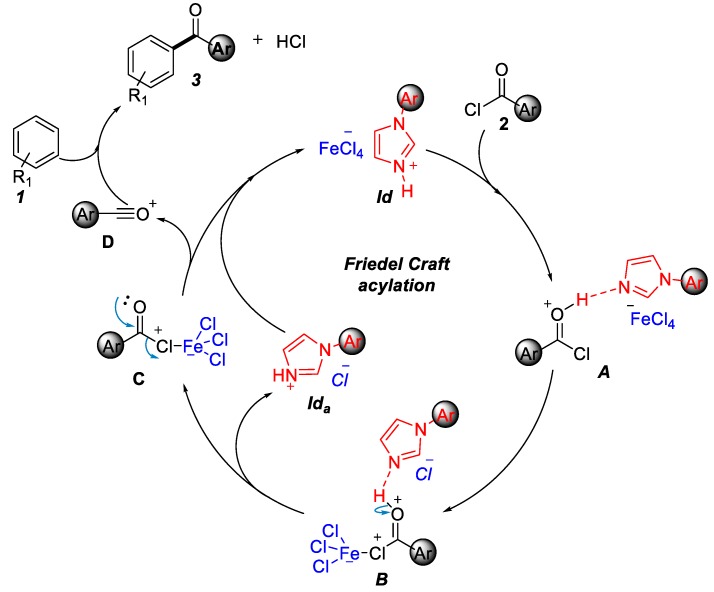
Proposed mechanism of Friedel–Crafts acylation.

**Figure 4 molecules-25-00352-f004:**
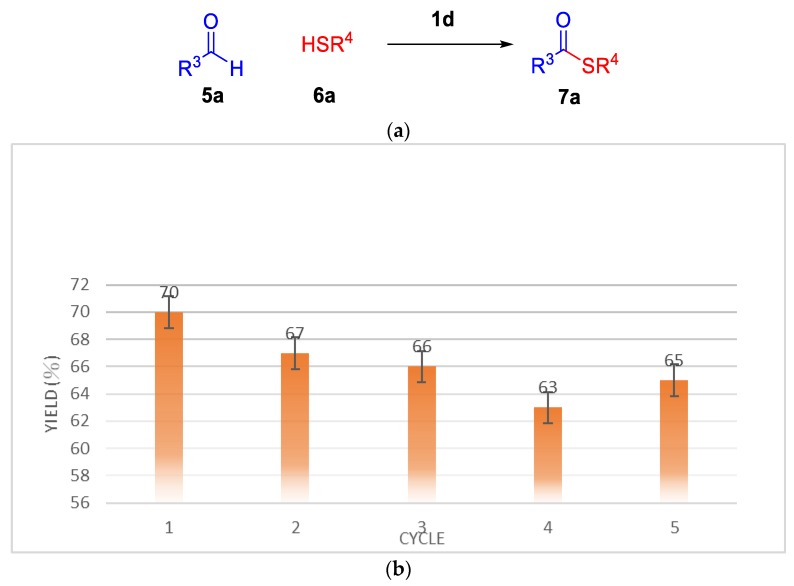
(**a**) Thioesterification reaction of thiols (**b**) Recycling of **ld** in the synthesis of S-Dodecyl benzothioate.

**Figure 5 molecules-25-00352-f005:**
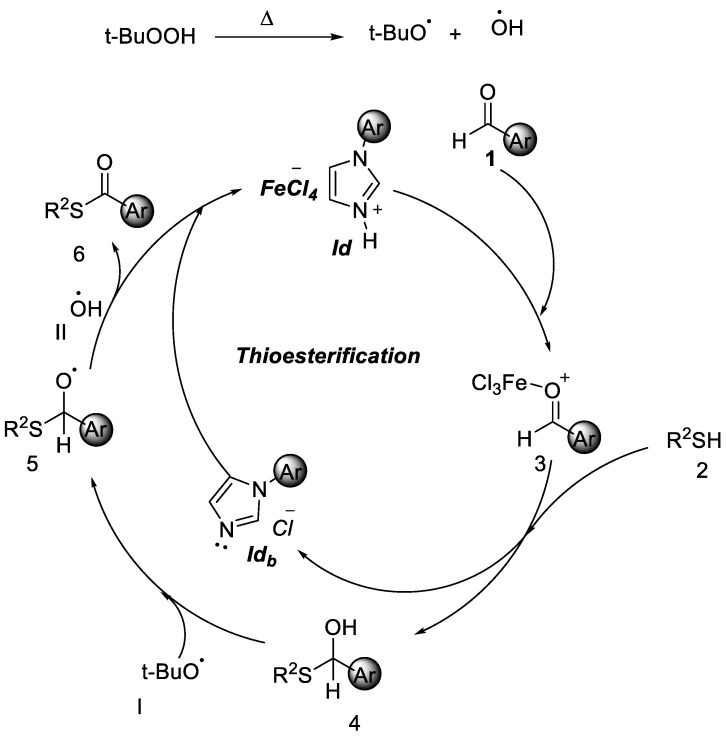
Proposed mechanism of thioesterification.

**Table 1 molecules-25-00352-t001:**
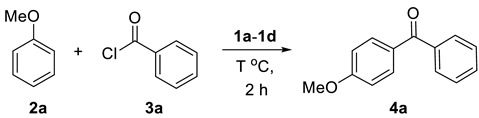
Optimized condition of Friedel–Crafts acylation.

Entry	IL (equiv.) *^a^*	T	Yield *^b^*
1	**1a** (1.0)	80 °C	57%
2	**1b** (1.0)	80 °C	32%
3	**1c** (1.0)	80 °C	72%
4	**1d** (1.0)	80 °C	74%
5	**1d** (1.0)	100 °C	74%
6	**1d** (1.0)	120 °C	71%
7	**1d** (0.9)	100 °C	78%
8	**1d** (0.8)	100 °C	73%

*^a^* Our reaction conditions are anisole (2.0 equiv.), benzoyl chloride (1.0 equiv.) and ionic liquids **1a**–**1d**. *^b^* The yields are isolated yields.

**Table 2 molecules-25-00352-t002:**
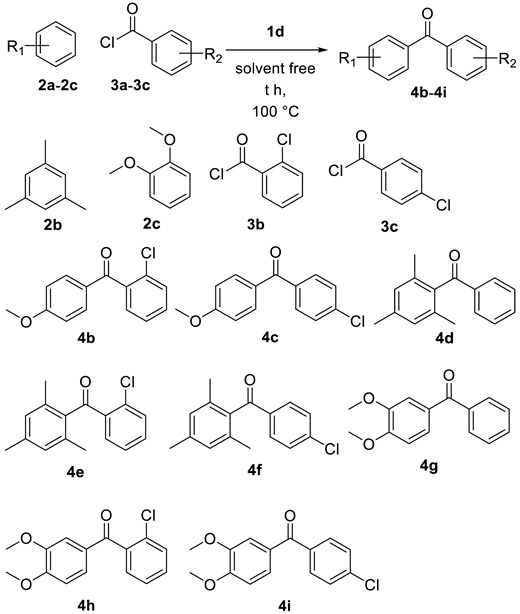
Ionic liquids 1d catalyzed Friedel—Crafts acylation with aryl alkanes **2a**–**2c** and acid chlorides **3a**–**3c** without solvent.

Entry	Aryl Alkane	Acyl Chloride	t (h)	P (yield)
1	**2a**	**3b**	4.0	**4b** (73%)
2	**2a**	**3c**	4.0	**4c** (68%)
3	**2b**	**3a**	2.0	**4d** (83%)
4	**2b**	**3b**	3.5	**4e** (79%)
5	**2b**	**3c**	2.0	**4f** (89%)
6	**2c**	**3a**	3.0	**4g** (81%)
7	**2c**	**3b**	3.5	**4h** (71%)
8	**2c**	**3c**	3.0	**4i** (70%)

**Table 3 molecules-25-00352-t003:**
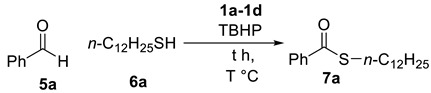
Optimized condition of thioesterification.

Entry	IL (equiv.) *^a^*	T	t (h)	Yield *^b^*
1	**1a** (0.025)	120 °C	2 h	33%
2	**1b** (0.025)	120 °C	2 h	43%
3	**1c** (0.025)	120 °C	2 h	56%
4	**1d** (0.025)	120 °C	2 h	70%
5	**1d** (0.025)	100 °C	2 h	68%
6	**1d** (0.025)	140 °C	2 h	69%
7	**1d** (0.010)	120 °C	2 h	68%
8	**1d** (0.030)	120 °C	2 h	61%

*^a^* Our reaction conditions are benzalehyde (5.0 equiv.), 1-dodecanethiol (1.0 equiv.), TBHP (2.0 equiv.) and ionic liquid **1a**–**1d**. *^b^* The yields are isolated yields.

**Table 4 molecules-25-00352-t004:**
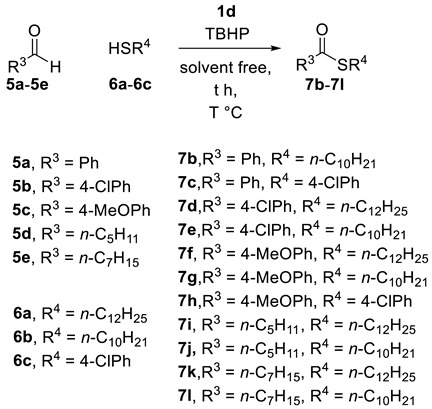
Ionic liquids **1d** catalyzed coupling reaction with aldehydes **5a**–**5e** and thiols **6a**–**6c** without solvent.

Entry	IL (eq) *^a^*	RCHO	Thiol	T	Yield *^b^*
1	**1d** (0.025)	**5a**	**6b**	120 °C	**7b** (88%)
2	**1d** (0.025)	**5a**	**6c**	120 °C	**7c** (55%)
3	**1d** (0.025)	**5b**	**6a**	120 °C	**7d** (73%)
4	**1d** (0.025)	**5b**	**6b**	120 °C	**7e** (73%)
5	**1d** (0.025)	**5c**	**6a**	120 °C	**7f** (76%)
6	**1d** (0.025)	**5c**	**6b**	120 °C	**7g** (73%)
7	**1d** (0.025)	**5c**	**6c**	120.°C	**7h** (51%)
8	**1d** (0.025)	**5d**	**6a**	120 °C	**7i** (80%)
9	**1d** (0.025)	**5d**	**6b**	120 °C	**7j** (71%)
10	**1d** (0.025)	**5e**	**6a**	120 °C	**7k** (73%)
11	**1d**(0.025)	**5e**	**6b**	120 °C	**7l** (64%)

*^a^* Our reaction conditions are aldehyde (5.0 equiv.), thiol (1.0 mmol), TBHP (2.0 equiv.) and ionic liquid **1d**. *^b^* The yields are isolated yields.

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
