# Peer review of "Tunable Aryl Imidazolium Recyclable Ionic Liquid with Dual Brønsted–Lewis Acid as Green Catalyst for Friedel–Crafts Acylation and Thioesterification"

_molecules, 2020, doi:10.3390/molecules25020352_

Round 1
Reviewer 1 Report
1. The authors didn't mention about ionic liquid synthesis and characterization details. They should consider this is important when it comes to catalysis and their reproducibilities.
2. In the reaction, how many times the authors carried about the same reaction. What is the standard deviation in the reaction?
3. Did the authors characterize the recycled catalyst? I believe it will change in the IL speciation. So, It's important to know the nature of recyclable catalysts after the reaction. The authors should consider discussing these points in the discussion.
Author Response
The authors didn't mention about ionic liquid synthesis and characterization details. They should consider this is important when it comes to catalysis and their reproducibilities.
Author Reply: Thank you for your valuable comments. Because one of our authors (Dr. Ho) mentioned about ionic liquid synthesis and characterization details in Materials and Applied Science, we also cited two references in our manuscript.
20 (a) Chang, J. C.; Yang, C. H.; Sun, I. W.; Ho, W. Y.; Wu, T.-Y. Synthesis and properties of magnetic aryl-imidazolium ionic liquids with dual Brønsted/Lewis acidity, Materials, 2018, 11, 2539. (b) Yang, C.H.; Chang, J.C.; Wu, T.Y.; Sun, I.W.; Wu, J.H.; Ho W.Y. Novel aryl-imidazolium ionic liquids with dual Brønsted/Lewis acidity as both solvents and catalysts for Friedel–Crafts alkylation, Sci. 2019, 9, 4743.
In the reaction, how many times the authors carried about the same reaction. What is the standard deviation in the reaction?
Author Reply: Thank you for your important comments. We did the same reaction more than 5 times and the yields of reaction are similar.
Did the authors characterize the recycled catalyst? I believe it will change in the IL speciation. So, it’s important to know the nature of recyclable catalysts after the reaction. The authors should consider discussing these points in the discussion.
Author Reply: We characterized the recycled catalysts by mass spectroscopy, and they show exactly same mass.
Reviewer 2 Report
In this manuscript, the authors report aryl imidazolium ionic liquid as a catalyst for FC acylation and thioesterification. As mentioned in the manuscript the catalyst seems robust and maintained activity after repeated use in subsequent reactions. The manuscript is well written and backed with adequate supplementary data.
Specific comments:
The introduction is lacking key citations regarding the previous work in the same field. For example, 1) Chem. Commun., 2005, 903-905; 2) Green Chem., 2002, 4, 129-133. Both of these papers used the ionic liquid in FC acylation reactions. The authors should cite the previous work in the field and comment on how their work contributes to the overall field. The substrate scope of the FC acylation reaction is not thoroughly studied. It is very important to show the limitation of the catalyst as well. A few key questions: do simple benzene or toluene will give any product formation? On the benzoyl chloride partner, electron-donating/withdrawing substituents tolerated? Table 3, the thioesterification reaction gave a similar yield at 100, 120, 140 °C, then why 120 °C was chosen to demonstrate substrate scope? Does lower than 100 °C have been tried?
Author Response
1.The introduction is lacking key citations regarding the previous work in the same field.
Author Reply: Thanks for your valuable comment. We added one sentence of ionic liquid on Friedel-Crafts acylation and also cited the references in the manuscript.
2. The substrate scope of the FC acylation reaction is not thoroughly studied. It is very important to show the limitation of the catalyst as well.
Author Reply: Thank you for your important comments. We studied three kind of substrates in this manuscript. Because the ionic liquid 1d exhibits high thermal stability, moisture insensitivity, recycling and reuse with minor loss in activity, the ionic liquid 1d also exhibited good solubility in many organic solvents and deionized water, but not in hexane.
3. Do simple benzene or toluene will give any product formation? On the benzoyl chloride partner, electron-donating/withdrawing substituents tolerated?
Author Reply: Yes. The desired products can be obtained by using benzene and toluene as our starting materials. However, because the boiling points of benzene and toluene are lower, molecular weights of their desired products are also small and compounds are easy to evaporate under vacuum. The electron-donating aryl alkane 1 (for example, R1 = CH3 and OCH3) reacted with acyl cation D easier than electron-withdrawing aryl alkane to form the desired product 3 in our proposed mechanism
4. Table 3, the thioesterification reaction gave a similar yield at 100, 120, 140 °C, then why 120 °C was chosen to demonstrate substrate scope? Does lower than 100 °C have been tried?
Author Reply: Thanks for your valuable comment. For thioesterification reaction, we found that the yield at 120 °C was the best. We tried to keep the reaction temperature at 80 °C, and the yield was only 61%
Please see the attachment.

Reviewer 3 Report
The manuscript submitted by Lin et al. to Molecules addresses synthesis of ionic liquids and their application for Friedel-Crafts acylation and thioesterfication. The research is well designed and conducted, and the manuscript is well written. In my opinion, it should be considered for publication in Molecules after considering the following issues:
Formation of HCl takes place during the Friedel-Crafts acylation. Did authors consider a possibility to remove it from the ionic liquid? Upon recycling, (page 6, lines 110-114) a gradual decrease of the yield takes place, but a 100% conversion is still observed. Why is that? What is responsible for a decrease of selectivity?
Author Response
- Formation of HCl takes place during the Friedel-Crafts acylation. Did authors consider a possibility to remove it from the ionic liquid?
Author Reply: Thank you for your important comment. Because HCl did not affect the Friedel-Crafts acylation, we did not consider to remove HCl from ionic liquid. It is possible to remove HCl from ionic liquid by using high vacuum or suction pump.
2. Upon recycling, (page 6, lines 110-114) a gradual decrease of the yield takes place, but a 100% conversion is still observed. Why is that? What is responsible for a decrease of selectivity?
Author Reply: Thank you for your important comment. When we monitored the reaction by TLC plate, we can see the desired product without any side product. We think that is 100% conversion. For the first run, the activity of ionic liquid 1d remained same yield (78%). In the subsequent second cycle (76%), the desired product was still reached with 100% conversion. Third (66%), fourth (69%), and fifth (67%) cycle, the yield slightly decreased. The loss of purification or extraction lead to decrease of yields.
Round 2
Reviewer 1 Report
I have consent for this manuscript for publication.
Author Response
Thanks for your acceptance
Reviewer 2 Report
The revised version looks good to me.
Author Response
Thanks for your valuable comments.
Reviewer 3 Report
The authors provide explanations in a point-by-point reply to reviewer, but did not add these explanations into the manuscript. In my opinion, the manuscript can be accepted for the publication after the text of the manuscript is accordingly modified and the explanations are given in the manuscript.
Author Response
the following changes have been applied:
line 102-104: While doing Friedel-Crafts acylation, HCl did not affect the reaction, so could not consider to remove HCl from ionic liquid. It is possible to remove HCl from an ionic liquid by using a high vacuum or suction pump.
line 119-120: When reaction was monitored on TLC plate, the desired product saw without any side product.